# Differential Diagnosis of Cachexia and Refractory Cachexia and the Impact of Appropriate Nutritional Intervention for Cachexia on Survival in Terminal Cancer Patients

**DOI:** 10.3390/nu13030915

**Published:** 2021-03-12

**Authors:** Nobuhisa Nakajima

**Affiliations:** Division of Community Medicine and International Medicine, University of the Ryukyus Hospital, Nishihara, Nakagami, Okinawa 903-0215, Japan; communit@w3.u-ryukyu.ac.jp or nakajy@med.u-ryukyu.ac.jp; Tel.: +81-98-895-1331

**Keywords:** palliative care, terminal cancer patients, cancer cachexia, refractory cachexia, nutritional intervention

## Abstract

Cancer cachexia subsequently shifts to refractory cachexia, however, it is not easy to properly differentiate them in clinical settings. Patients considered refractory cachexia may include cachectic patients with starvation. This study aimed to identify these cachectic patients and to evaluate the effect of nutritional intervention for them. Study subjects were terminal cancer patients admitted for palliative care and were judged refractory cachexia in the last five years. We retrospectively examined to find useful indices for identifying such cachectic patients and for evaluating the effect of nutritional intervention. Out of 223 patients in refractory cachexia, 26 were diagnosed cachexia with starvation after symptom management. Comparing before and one week after this management, Palliative Performance Scale (PPS) and transthyretin significantly improved (*p* < 0.0001, *p* = 0.0002, respectively) Then, we started nutritional intervention for these cachectic patients and divided into effective group (*n* = 17) and non-effective group (*n* = 9) using the criteria for cachexia. Comparing between the two groups, PPS significantly improved2 weeks after intervention in effective group (*p* = 0.006). Survival time was significantly longer in effective group (*p* = 0.008). PPS and transthyretin were useful for differential diagnosis of cachexia and refractory cachexia. PPS was useful for evaluating nutritional intervention for cachectic patients. Appropriate nutritional intervention improved survival.

## 1. Introduction

Cancer cachexia is a complex metabolic disorder characterized by progressive skeletal muscle loss (with or without fat loss), which is difficult to improve with conventional nutritional therapy and leads to various functional disorders [1]. Pathophysiologically, it is characterized by protein catabolism and energy imbalance caused by a variety of interplays between anorexia and metabolic disorders [2,3]. Cancer cachexia is a complex multifactorial condition caused by tumor-host interaction, and its main component is chronic systemic inflammation [1,4].

Cancer cachexia is classified into the following three stages: pre-cachexia, cachexia, and refractory cachexia [5]. Cachexia is defined as weight loss of >5% over the past 6 months, often reduced food intake and systemic inflammation. Refractory cachexia is defined as a cancer disease both procatabolic and not responsive to anticancer treatment, with low performance status (World Health Organization score 3–4) and life expectancy of <3 months. Cancer cachexia subsequently shifts to refractory cachexia as the disease progresses, and nutritional intervention is rarely indicated [6].

By the way, it is not easy to properly differentiate between cachexia and refractory cachexia in clinical settings. Therefore, patients considered to be suffering from refractory cachexia may include cachectic patients who have fallen into starvation due to poor symptom management and inappropriate nutritional support. If we can identify these cachectic patients with starvation and provide them with appropriate nutritional support, their general condition and prognosis may be improved.

The purpose of this study was (1) to identify cachectic terminal cancer patients with starvation among the patients who were judged to be suffering from refractory cachexia, and (2) to evaluate the impact of appropriate nutritional intervention on the survival of these cachectic patients with starvation.

## 2. Materials and Methods

Study subjects were terminal cancer patients who were admitted for palliative care and were judged to have refractory cachexia due to severe malnutrition and poor general condition in the last five years (Higashisapporo hospital: April 2012–March 2014; and Tohoku University hospital: April 2014–March 2017). We retrospectively examined the following two methods.

### 2.1. Identification of Useful Indices for Identifying Starveling Cachectic Patients Who May Be Eligible for Nutritional Treatment among Patients Considered to Have Refractory Cachexia

Patients who met the criteria for refractory cachexia based on the criteria for cachexia [5] at the time of admission were selected. These patients underwent the necessary interventions, such as symptom management including pain management and peripheral hydration therapy to improve dehydration, if needed. One week after these interventions, we again differentiated between patients with refractory cachexia and cachectic patients with starvation (i.e., patients who were considered to have refractory cachexia at the time of admission, but underwent appropriate symptom relief and hydration, and were subsequently diagnosed as possibly in the cachexia stage) using the criteria for cachexia. We used Palliative Performance Scale (PPS) [7] and Palliative Performance Index (PPI) [8] for the evaluation of performance status and survival time, which were the diagnostic criteria for refractory cachexia. PPS [7] and PPI [8] were measured for cachectic patients with starvation. In addition, blood biochemical tests (blood count and blood fractions, albumin, transthyretin (TTR) and C-reactive protein (CRP)) were performed; based on them, the Controlling Nutritional Status score (CONUT score) [9], Glasgow Prognostic Scale (GPS) [10], and neutrophil-to-lymphocyte ratio (NLR) [11] were determined. CONUT score consists of the following three items: albumin, total lymphocyte count and cholesterol, and is classified into the following three stages: mild malnutrition, moderate malnutrition and severe malnutrition [9]. These data were compared before (at the admission) and one week after the intervention.

The selection of the above items was based on the items to classify the stage of cachexia, the fact that the pathophysiology of cachexia is protein catabolism and energy imbalance, and the fact that the main component of cachexia is chronic systemic inflammation.

### 2.2. Identification of Useful Indices to Evaluate the Effect of Nutritional Intervention on Cachectic Patients with Starvation

Patients with cachexia associated with starvation underwent artificial hydration and nutrition therapy (AHNT) based on the guideline for parenteral fluid management in terminal cancer patients (edited by the Japanese Society for Palliative Medicine; JSPM) [12]. As these 26 patients exhibited gastrointestinal symptoms, such as nausea, vomiting, and appetite loss, AHNT was performed instead of oral nutrition or enteral nutrition via the nasogastric tube. Total parenteral nutrition was mainly performed as AHNT, and when securing a central venous route was difficult, peripheral parenteral nutrition (containing 7.5% glucose and 3% amino acids) was used. Blood biochemical tests including transthyretin were regularly performed to properly evaluate the effectiveness of AHNT. In the subsequent course of the study, based on the diagnostic criteria for cachexia, we determined whether the patients were maintaining the cachexia stage (i.e., the nutritional intervention was effective) or shifting to the refractory cachexia stage (i.e., the nutritional intervention was non-effective).

Considering the mean survival time of patients with refractory cachexia, we defined the effective group as those who maintained the stage of cachexia for more than three weeks after the start of the nutritional intervention, and the non-effective group as those who shifted to the stage of refractory cachexia within three weeks [5,8,12,13]. The indices described in 2.1 were measured every week before and after the nutritional intervention, and compared week by week between the two groups. In addition, the survival time was compared between the two groups.

### 2.3. Guideline for Parenteral Fluid Management in Terminal Cancer Patients (GL)

The primary purpose of this guideline was to aid clinicians in making a clinical decision regarding AHNT in order to ensure a better quality of care for terminal cancer patients [12]. The target population was that of adult patients with incurable cancer whose remaining life expectancy was one month or less (i.e., the time until cachexia was expected to substantially impair physiological functions), who were not currently receiving anticancer treatment, and were unable to take sufficient fluids and food orally, despite the appropriate support. This guideline was constructed according to evidence-based and formal consensus-building methods using the Delphi technique. It included general and specific recommendations. The specific recommendations included 25 recommendations on physical suffering (e.g., the general quality of life, ascites, nausea and vomiting, thirst, pleural effusion, tracheal secretions, delirium, malaise, and edema) and the remaining life expectancy; 10 nursing-related recommendations; and 7 ethical recommendations. In addition, this guideline recommended total parenteral nutrition for patients whose nutritional status or general condition could be either maintained or improved with this support and for patients with pre-cachexia or cachexia but not refractory cachexia [5].

Recently, many academic societies have actively developed clinical practice guidelines. However, some variation exists in the development process as well as the quality of the product. The Japan Council for Quality Health Care (JQ) manages guidelines in Japan. The JQ evaluated the quality of 519 guidelines (87 for cancer and 432 for noncancer) published from 2011 to 2017 using the appraisal of guideline for research and evaluation II (AGREE II) [14]. As a result, the quality of cancer-related guidelines was higher than that of noncancer guidelines. According to the evaluation of Domain 3 (Rigor of Development) of AGREE II, 10 guidelines had a score of >80 among cancer-related guidelines, and 5 of them were guidelines on palliative care published by JSPM; this guideline on AHNT had the highest score (87 points) [15]. This meant that this guideline had a methodologically high quality on hydration therapy.

### 2.4. Statistical Analysis

The Wilcoxon signed-ranks test was used to assess changes of indices according to symptom management and hydration. The Mann-Whitney U test was used to compare data after nutritional intervention. Differences resulting in *p*-values of <0.05 were considered statistically significant. All analyses were performed using the JMP15 software (Cary, NC, USA).

## 3. Results

At the time of admission, 223 patients were considered to have refractory cachexia based on the criteria for cachexia [5]. Among them, 26 patients were diagnosed cachexia with starvation after appropriate symptom management and fluid administration by evaluating the stage of cachexia again based on the criteria for cachexia [5]. The summary of these 26 cases at the time of admission was shown in Table 1. Symptom management, including pain and gastrointestinal symptoms, was generally poor. PPS was ≤30 in all patients. PPI was >6 pts in all patients, and this meant that the expected prognosis was less than three weeks. Blood biochemical tests revealed albumin <2.5 g/dL in all patients and TTR ≤10 mg/dL in 90% of patients. The CONUT score was ≥9 pts in 70% of cases, indicating severe nutritional disorders. GPS was 2 pts in 80% of cases, and NLR was >4 in 80% of cases. Comparing the above before and one week after intervention, PPS was 20 (20–30) vs. 60 (50–60) and TTR was 9 (8–10) vs. 13 (11–18) with significant improvement (*p* < 0.0001, *p* = 0.0002, respectively). Albumin did not change significantly between pre-intervention and one week after intervention (Table 2).

AHNT based on the GL [12] was initiated in cachectic patients with starvation, and albumin, TTR and PPS were compared week by week between the effective group (*n* = 17) and the non-effective group (*n* = 9), and PPS showed a significant difference (60 (60–60) vs. 50 (50–50); *p* = 0.01) after two weeks of intervention. Albumin and TTR did not differ significantly after two weeks of intervention, and a significant difference was observed between the two groups after three weeks of intervention (albumin: 2.9 (2.8–3.0) vs. 2.5 (2.5–2.7); *p* = 0.03, TTR: 24 (22–25) vs. 18 (16–19); *p* = 0.008). Survival time was significantly longer in effective group (52 days (46–60) vs. 23 days (19–27), *p* = 0.008) (Table 3).

## 4. Discussion

AHNT has been proposed for advanced cancer patients with cachexia who are unable to ensure adequate nutrition by oral intake or enteral nutrition. AHNT has been shown to be effective at home for patients with advanced cancer who have a prognosis of more than 2–3 weeks, and is recommended for chronic impaired nutritional intake. On the other hand, there are recommendations to refrain from AHNT for patients with a shorter prognosis and those facing imminent death [16]. Although many patients whose cachexia shifted to refractory cachexia fall into this category and nutritional intervention is rarely indicated, it is not easy to properly differentiate between cachexia with starvation and refractory cachexia, which is the topic of this study. Therefore, we used PPS [7], PPI [8], albumin, TTR, CONUT score [9], GPS [10], and NLR [11] as candidate indices for this purpose, and attempted to identify appropriate indicators among them.

We also aimed to identify indicators that may be useful as a basis for judging the effectiveness/non-effectiveness of nutritional interventions, i.e., indicators that show significant changes when the stage shifts from cachexia to refractory cachexia, and to clarify the timing of such changes.

PPS, a modification of the Karnofsky Performance Scale, is presented as a new tool for the assessment of physical status in palliative care [7]. This scale is shown in Table 4. Physical performance is divided into 11 categories, measured in 10% decremental levels, from fully ambulatory and healthy (100%) to death (0%). The factors which differentiate these levels are based on five observable parameters: the degree of ambulation, ability to perform activities/extent of disease, ability to carry out self-care, food/fluid intake, and state of consciousness. In this study, PPS was shown to be one of the useful indices in differentiating cachexia with starvation from those with refractory cachexia. Furthermore, PPS was found to be the most rapidly changing indicator in assessing whether or not nutritional intervention was effective.

PPI was defined by the following five items: performance status (PPS), oral intake, edema, dyspnea at rest, and delirium [8]. When a PPI of more than 6 was used as a cutoff point, the three-week survival was predicted with a sensitivity of 80% and specificity of 85%. When a PPI of more than 4 was used as a cutoff point, the six-week survival was predicted with a sensitivity of 80% and specificity of 77%. In conclusion, whether patients live longer than three or six weeks can be acceptably predicted by PPI. In this study, it was thought that we were able to make a generally valid assessment in predicting the prognosis of the group that did not respond to nutritional intervention.

TTR is one of the rapid-turnover proteins with a shorter half-life (2 days) compared with albumin (20 days) [17] and is well-known as a nutritional marker [18]. Additionally, TTR might be an indicator of systemic inflammation, because systemic inflammation might have the possibility of reducing the transcription of TTR via the decreased binding of hepatic nuclear factor-4 alpha to the promoter of the TTR gene in hepatocyte [17,19]. A rat model showed that TTR levels decreased after 14 days of consuming a diet that contained only 60% of required proteins [20]. Adequate nutrition was shown to increase TTR levels to the normal levels within 4–8 days in malnourished children [21]. Therefore, TTR might rapidly reflect the nutritional status and systemic inflammation, and low TTR levels was an indicator of poor prognosis among cancer patients in palliative care settings [22,23]. In this study, TTR, like PPS mentioned above, was shown to be one of the useful indices in differentiating cachectic patients with starvation from those with refractory cachexia. However, it did not change as quickly as PPS in determining whether nutritional intervention was effective.

GPS is a scoring system that combines CRP and albumin, and is a prognostic score for cancer patients independent of the stage of the disease [10]. The cutoff values are 1.0 mg/dL for CRP and 3.5 g/dl for albumin, with CRP ≥1.0 mg/dl and albumin <3.5 g/dL (=GPS 2) having the poorest prognosis. GPS is also useful for advanced cancer patients in palliative care (palliative care unit, palliative care team in general wards, home palliative care), and 71% of the patients had CRP ≥1.0 mg/dL and albumin <3.5 g/dL (=GPS 2) [24]. In our study, we found that 80% of the patients had CRP ≥1.0 mg/dL and albumin <3.5 g/dL (=GPS 2). In this study, this criterion was met in 80% of the patients at the time of admission; even when nutritional intervention resulted in improvements in other measures, the GPS did not change. Therefore, GPS was not a useful index to assess the effectiveness of nutritional intervention.

NLR is a measure of neutrophil and lymphoid cells, and a value of 4 or higher indicates poor prognosis [11]. In this study, 80% of patients had NLR ≥4, and even when nutritional intervention resulted in improvement in other measures, it did not change, as GPS did. Therefore, NLR was not also a useful index in assessing the effectiveness of nutritional intervention.

Although the effect of nutritional support considering the pathophysiology of cachexia in advanced cancer patients is promising, there is still insufficient evidence and further research progress is required [25]. Under such circumstances, this study revealed that PPS and TTR are useful for the differential diagnosis of cachexia/refractory cachexia in the terminal stage of cancer and for determining the indications for nutritional treatment, and that PPS is the most rapid and useful for determining the appropriateness of continuing nutritional intervention for cachectic patients. 

Several studies have demonstrated that cancer cachexia patients subjected to aggressive refeeding can develop an overfeeding reaction during the first 2–3 weeks [26,27]. Therefore, it is important to perform AHNT under strict medical control in order to avoid undesirable reactions. Appropriate nutritional interventions based on these findings are expected to improve the general condition and prolong the survival of patients with cancer cachexia, thereby improving the quality of care for terminal cancer patients.

There are limitations to this study, including the fact that it is a retrospective observational study with a small sample size. A multicenter interventional study based on the findings of this study would be required.

## 5. Conclusions

It is important to identify cachectic terminal cancer patients with starvation among patients having refractory cachexia, and appropriate nutritional intervention based on the findings of this study may lead to a better general condition and prognosis.

## Figures and Tables

**Table 1 nutrients-13-00915-t001:** Patients’ characteristics on admission.

	Diagnosis	Pain	A/L	N/V	Fatigue	Alb	TTR	PPS	PPI	CONUT	GPS	NLR	TPN/PPN
1	Stomach	++	++	+	++	1.8	10	20	7.5	11	2	7.2	TPN
2	Pancreases	++	+	+	++	2.1	8	20	11	9	2	6.4	TPN
3	Pancreases	+	+	-	+	2.2	10	20	7.5	7	1	3.3	PPN
4	Stomach	++	++	++	+	2.0	9	20	10	10	2	7.7	TPN
5	Stomach	++	++	+	+	2.0	6	20	6.5	9	2	6.3	TPN
6	Lung	+	+	-	+	2.3	12	30	9.5	10	2	8.5	PPN
7	Colon	+	+	-	+	2.4	8	30	9.5	8	1	3.5	TPN
8	Stomach	++	++	+	++	2.0	10	20	7.5	10	2	5.8	TPN
9	Stomach	++	+	+	++	2.2	9	20	11	10	2	8.2	TPN
10	Pancreases	++	++	+	+	2.0	11	20	10	11	2	7.7	PPN
11	Bile duct	++	++	+	+	2.0	9	30	7.5	9	2	5.1	TPN
12	Rectum	+	+	+	+	2.4	8	30	7.5	8	2	4.6	TPN
13	Lung	+	+	-	+	2.2	10	30	9.5	8	1	3.5	TPN
14	Liver	++	++	+	++	2.3	8	20	7.5	10	2	8.0	TPN
15	Lung	++	++	+	++	2.1	10	20	7.5	9	2	6.0	TPN
16	Colon	+	+	-	+	2.3	10	30	6.5	8	2	8.0	TPN
17	Stomach	++	++	++	+	2.0	9	20	10	10	2	7.6	TPN
18	Stomach	++	++	++	+	2.0	9	20	6.5	8	2	5.6	TPN
19	Lung	++	++	+	++	2.3	8	20	7.5	9	2	8.8	TPN
20	Lung	++	++	+	+	2.2	8	30	9.5	10	2	5.2	PPN
21	Colon	+	+	-	+	2.1	10	30	9.5	9	1	3.4	TPN
22	Pancreases	++	++	+	+	1.9	8	30	8.5	10	2	4.6	TPN
23	Colon	+	+	+	+	2.3	10	30	6.5	8	2	4.3	TPN
24	Stomach	++	++	+	+	2.0	8	20	7.5	8	2	6.5	TPN
25	Pancreases	++	++	+	+	2.0	9	30	7.5	8	2	5.6	TPN
26	Stomach	++	++	+	+	1.8	12	20	10	11	2	7.2	TPN

A/L: Appetite Loss, N/V: Nausea/Vomiting, PPS: Palliative Performance Scale, PPI: Palliative Prognostic Index, GPS: Glasgow Prognostic Score, NLR: Neutrophil-to-Lymphocyte Ratio.

**Table 2 nutrients-13-00915-t002:** Changes of indices by symptom management and hydration.

	Before Median (Interquartile Range)	1 Week after Median (Interquartile Range)	*p*-Value
PPS	20 (20–30)	60 (60–60)	<0.0001
TTR (mg/dL)	9 (8–10)	13 (9–16)	0.0002
albumin (g/dL)	2.1 (2.0–2.3)	2.2 (2.0–2.4)	0.19
CONUT score	9 (8–10)	9 (8–10)	0.18
GPS	2 (2–2)	2 (2–2)	0.71
NLR	6.2 (4.6–7.7)	5.9 (4.3–7.0)	0.18

PPS: Palliative Performance Scale, TTR: Transthyretin, CONUT score: Controlling Nutritional Status score, GPS: Glasgow Prognostic Scale, NLR: Neutrophil to Lymphocyte ratio.

**Table 3 nutrients-13-00915-t003:** Comparison of the data after nutritional intervention.

Items		Effective Group Median (Interquartile Range)	Effective Group Median (Interquartile Range)	*p*-Value
PPS	before	20 (20–30)	30 (20–30)	0.10
	1 week	60 (60–60)	60 (50–60)	0.13
	2 weeks	60 (60–60)	50 (50–50)	0.006
	3 weeks	60 (60–60)	40 (40–40)	0.006
TTR (mg/dL)	before	9 (8–10)	9 (8–10)	>0.9999
	1 week	13 (9–16)	14 (11–17)	0.34
	2 weeks	20 (18–23)	19 (18–23)	0.07
	3 weeks	24 (22–25)	18 (16–19)	0.008
Albumin (g/dL)	before	2.1 (2.0–2.3)	2.0 (1.9–2.2)	0.67
	1 week	2.2 (2.1–2.5)	2.2 (2.0–2.3)	0.67
	2 weeks	2.6 (2.4–2.8)	2.4 (2.3–2.5)	0.18
	3 weeks	2.9 (2.8–3.0)	2.5 (2.5–2.7)	0.03
Survival periods (days)		52 (46–60)	23 (19–27)	0.008

Abbreviations: PPS: Palliative Performance Scale, TTR: Transthyretin.

**Table 4 nutrients-13-00915-t004:** Palliative Performance Scale (PPS).

%	Ambulation	Activity and Evidence of Disease	Self-Care	Intake	Conscious Level
100	Full	Normal activity No evidence of disease	Full	Normal	Full
90	Full	Normal activity Some evidence of disease	Full	Normal	Full
80	Full	Normal activity with effort Some evidence of disease	Full	Normal or Reduced	Full
70	Reduced	Unable normal job/work Some evidence of disease	Full	As above	Full
60	Reduced	Unable hobby/hose work Significant disease	Occasional assistance necessary	As above	Full or Confusion
50	Mainly Sit/Lie	Unable to do any work extensive disease	Considerable assistance required	As above	As above
40	Mainly Bed Bound	As above	Mainly assistance	As above	Full or Drowsy or Confusion
30	Totally Bed Bound	As above	Total care	Reduced	As above
20	As above	As above	As above	Minimal sips	As above
10	As above	As above	As above	Mouth care only	Drowsy or Coma

## Data Availability

Data described in the manuscript will be made available to qualified researchers upon reasonable request.

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
