# Peer review of "Differential Diagnosis of Cachexia and Refractory Cachexia and the Impact of Appropriate Nutritional Intervention for Cachexia on Survival in Terminal Cancer Patients"

_nutrients, 2021, doi:10.3390/nu13030915_

Round 1

Reviewer 1 Report

It is a great pleasure for me to have the opportunity to review your paper.

The theme of this paper is informative.

I have several minor comments as follows.

I am afraid that the description of the methods and results in this paper may have been a bit complicated and difficult to understand.

1) in 2.1.3, “These data were compared before and one week after the intervention.” The timeline is confusing.

>>> These data were compared before (at the admission) and one week after the intervention.

2) in .2.2.1, “Patients with cachexia associated with starvation” is written, but how about unifying it with the writing "starveling cachectic patients" as written in 2.1.2?

3) in 2 (materials and methods), I suggest you add a patient flow diagram.

4) in 3 (results), I suggest you add subtitles to make it easier to see that each result also accommodate each method 2.1 and 2.2.

5) The statistical analysis was not well detailed out.

Author Response

Response to Reviewers’ comments [Reviewer 1]

Thank you for reviewing our paper.

We revised the manuscript according to the reviewer’s comments.

The changes are shown in red and underlined.

Q-1

in 2.1.3, “These data were compared before and one week after the intervention.” The timeline is confusing.>>> These data were compared before (at the admission) and one week after the intervention.

A-1

According to the reviewer’s comment, we revised this part as follows:

These data were compared before (at the admission) and one week after the intervention.

Q-2

in .2.2.1, “Patients with cachexia associated with starvation” is written, but how about unifying it with the writing "starveling cachectic patients" as written in 2.1.2?

A-2

As the reviewer pointed out, “patients with cachexia associated with starvation” and “starveling cachectic patients” have the same meaning.

First, we explained “patients with cachexia associated with starvation”, and after that, we unified the notation in “starveling cachectic patients”

Q-3

in 2 (materials and methods), I suggest you add a patient flow diagram.

A-3

It may be useful to present such a flow diagram in a prospective study, but in this study, we refrained from using a flow diagram in “Materials and Methods” as the number of cases shown here was obtained as retrospective data.

Q-4

in 3 (results), I suggest you add subtitles to make it easier to see that each result also accommodate each method 2.1 and 2.2.

A-4

Based on the reviewer’s comment, we revised the “Results” section to make easier to understand by arranging tables and displaying the title of tables.

Q-5

The statistical analysis was not well detailed out.

A-5

Based on the reviewer’s comment, we revised this part in the “Statistical analysis” section as follows:

The Wilcoxon signed-ranks test was used to assess changes of indices according to symptom management and hydration. The Mann-Whitney U test was used to compare data after nutritional intervention. Differences resulting in p-values of <0.05 were considered statistically significant. All analyses were performed using the JMP15 software (Cary, NC, USA)..

Reviewer 2 Report

The manuscript titled: “Differential diagnosis of cachexia and refractory cachexia and the impact of appropriate nutritional intervention for cachexia on survival in terminal cancer patients” appears of some interest.   

As suggested by the Author, it is not easy to differentiate between cachexia and refractory cachexia in clinical settings.

Overall, the paper is well written. However, some limitations are present and my comments are the following:

  1. I would specify in the section of “Materials and Methods” the reason to choose non-parametric statistic approach. Furthermore, it is no clear the reason Author performed non-parametric tests to analyse data, when scores of scales and indexes considered are presented as mean±standard deviation (Table 2).
  2. I am not agree with Authors about the choice of statistical tests. In particular, in Table 2, differences between “before” and “1 week after” are repeated measures on the same subjects. I would suggest to perform the Wilcoxon signed-rank test instead of the Mann-Withney U test, which is recommended for independent measures. I would suggest to use the same approach for multiple comparisons. Also in this case, I would use the Friedman test, the non-parametric test for repeated measures.

Author Response

Response to Reviewers’ comments [Reviewer 2]

Thank you for reviewing our paper.

We revised the manuscript according to the reviewer’s comments.

The changes are shown in red and underlined.

Q-1

I would specify in the section of “Materials and Methods” the reason to choose non-parametric statistic approach. Furthermore, it is no clear the reason Author performed non-parametric tests to analyse data, when scores of scales and indexes considered are presented as mean±standard deviation (Table 2).

Q-2

I am not agree with Authors about the choice of statistical tests. In particular, in Table 2, differences between “before” and “1 week after” are repeated measures on the same subjects. I would suggest to perform the Wilcoxon signed-rank test instead of the Mann-Withney U test, which is recommended for independent measures. I would suggest to use the same approach for multiple comparisons. Also in this case, I would use the Friedman test, the non-parametric test for repeated measures.

A-1 and 2

Thank you for your important comments regarding statistical analysis.

We think that it is appropriate to use Wilcoxon signed-ranks test for assessing the changes of indices by symptom management and hydration, so we revised in the the “Statistical analysis” section as follows:

The Wilcoxon signed-ranks test was used to assess changes of indices according to symptom management and hydration. The Mann-Whitney U test was used to compare data after nutritional intervention. Differences resulting in p-values of <0.05 were considered statistically significant. All analyses were performed using the JMP15 software (Cary, NC, USA)..

Reviewer 3 Report

I’m surprised that such an extensive work has been done by a single author, as detailed in the paper to revise. In my opinion, all the people engaged in the study in collecting and analyzing samples shall be included in the author list. In this regard, in the section Ethics, the authour says We… so I believe a team work is listed and the other people involved shall be listed as authour as well.

Minor concerns are about some words used for example starveling seems improver to me as employed , please find another word.

In the Discussion session, please explain better the sentence

On the other hand, there are recommendations to refrain from AHNT for patients with a shorter prognosis and those facing imminent death

Discuss also about overfeeding reaction in terminal cachectic patients (see paper Aquila, Re Cecconi et al., 2020 Cells).

For the rest, I think the paper is well written and the conclusions correctly drawn.

Author Response

Thank you for your reviewing our paper. I revised the manuscript according to the reviewer’s comments. The changes are shown in red and underlined.

Q-1: I’m surprised that such an extensive work has been done by a single author, as detailed in the paper to revise. In my opinion, all the people engaged in the study in collecting and analyzing samples shall be included in the author list. In this regard, in the section Ethics, the authour says We… so I believe a team work is listed and the other people involved shall be listed as author as well.

A-1: According to the reviewer’s comment, two physicians who cooperated with this study were listed in the “Acknowledgment” section.  (In fact, I wanted to co-author a physician who contributed most to carrying out this study, but he sadly died of a malignant neoplasm two years ago. Then, I could not add his name to the co-author.)

Acknowledgment

The author would like to express the gratitude to Kazuhiko Koike M.D., Ph.D. and Yuji Takahashi M.D., Ph.D. for creating study design and data analysis.

Q-2: Minor concerns are about some words used for example starveling seems improver to me as employed, please find another word.

Q-3: In the Discussion session, please explain better the sentence

A-2,3: In accordance with the reviewer’s comment, we revised the terms such as unifying starveling cachectic patients to cachectic patients with starvation in the “Materials and Methods”, “Results” and “Discussion” sections.

Q-4: On the other hand, there are recommendations to refrain from AHNT for patients with a shorter prognosis and those facing imminent death

A-4: I basically agree with this comment. However, it is also true that there are small number of cachectic patients with shorter prognosis for whom AHNT is indicated. This study focused on these patients.

Q-5: Discuss also about overfeeding reaction in terminal cachectic patients (see paper Aquila, Re Cecconi et al., 2020 Cells).

A-5: Thank you very much for the valuable comment. I added the following sentence to the “Discussion” section with reference to the introduced paper.

Several studies have demonstrated that cancer cachexia patients subjected to aggressive refeeding can develop an overfeeding reaction during the first 2-3 weeks. Therefore, it is important to perform AHNT under strict medical control in order to avoid undesirable reactions.

Reviewer 4 Report

Authors try to identify starveling cachectic patients and to evaluate the effect of nutritional intervention for them. Study subjects were terminal cancer patients admitted for palliative care and were judged refractory cachexia in the last five years. Authors retrospectively examined to find useful indices for identifying starveling cachectic patients and for evaluating nutritional intervention using Palliative Performance Scale (PPS) and transthyretin, and finally, authors identified PPS and transthyretin significantly improved (p=0.010, 0.012, respectively), PPS and transthyretin were useful for differential diagnosis of cachexia and refractory cachexia. PPS was useful for evaluating nutritional intervention for cachectic patients. Appropriate nutritional intervention improved survival. Some concerns were raised.

  1. The manuscript is hard to understand easily. Authors have to identify and document these definitions such as “refractory cachexia “, “starveling cachectic patients”, “Controlling Nutritional Status score (CONUT score) “and so on. The definition is different in different studies and hard to read by our readers.
  2. How to perform “artificial hydration and nutrition therapy (AHNT), it is not clear
  3. In result section, “Of these, twenty-six cases were judged to have cachexia with starvation after appropriate symptom management and administration of fluids”, How to define “cachexia with starvation”, it is still not clear.
  4. Majority of these refractory cachexia received TPN but some received PPN, reasons? Any patients received TPN or PPN, but nasogastric feeding or oral intake is still the most common nutrition supply method for these terminal cancer patients worldwide. How to applicate these methods to other countries?
  5. The study is retrospective or prospective study? Authors said that “we started nutritional intervention for starveling cachectic patients and divided into effective group (n=17) and non-effective group (n=9) using the criteria for cachexia”, a prospective study?
  6. IRB is lacking.

Author Response

Response to Reviewers’ comments [Reviewer 3]

Thank you for reviewing our paper.

We revised the manuscript according to the reviewer’s comments.

The changes are shown in red and underlined.

Q-1

The manuscript is hard to understand easily. Authors have to identify and document these definitions such as “refractory cachexia “, “starveling cachectic patients”, “Controlling Nutritional Status score (CONUT score) “and so on. The definition is different in different studies and hard to read by our readers.

A-1

Based on the reviewer’s comment, we revised the explanation of “refractory cachexia”, “starveling cachectic patients” and “CONUT score” in the “Introduction” and “Materials and Methods” sections.

Cachexia is defined as weight loss of >5% over the past 6 months, often reduced food intake and systemic inflammation. Refractory cachexia is defined as a cancer disease both procatabolic and not responsive to anticancer treatment, with low performance status (WHO score 3-4) and life expectancy of <3 months.

Starveling cachectic patients are the patients with cachexia who have fallen into starvation due to poor symptom management and inappropriate nutritional support.

CONUT score consists of the following three items: albumin, total lymphocyte count and cholesterol, and is classified into the following three stages: mild malnutrition, moderate malnutrition and severe malnutrition.

Q-2

How to perform “artificial hydration and nutrition therapy (AHNT), it is not clear

Q-4

Majority of these refractory cachexia received TPN but some received PPN, reasons? Any patients received TPN or PPN, but nasogastric feeding or oral intake is still the most common nutrition supply method for these terminal cancer patients worldwide. How to applicate these methods to other countries?

A-2 and 4

As the reviewer pointed out, nasogastric feeding or oral intake is the most common nutrition supply method for terminal cancer patients not only in the world but also in Japan.

The patients in this study had gastrointestinal symptoms such as nausea and vomiting, so we performed nutritional treatment via intravenous route instead of enteral nutrition.

We added the explanation in the “Materials and Methods” section as follows:

As these patients exhibited gastrointestinal symptoms such as nausea and vomiting, AHNT was performed instead of enteral nutrition via the nasogastric tube. Total parenteral nutrition was mainly performed as AHNT, and when securing a central venous route was difficult, peripheral parenteral nutrition (containing 7.5% glucose and 3% amino acids) was used.

Q-3

In result section, “Of these, twenty-six cases were judged to have cachexia with starvation after appropriate symptom management and administration of fluids”, How to define “cachexia with starvation”, it is still not clear.

A-3

After appropriate symptom management and administration of fluids, we evaluated the stage of cachexia in 223 patients again based on the criteria for cachexia [5]. We revised the relevant part in the “Results” section as follows:

Among them, 26 patients were diagnosed cachexia with starvation after appropriate symptom management and fluid administration by evaluating the stage of cachexia again based on the criteria for cachexia [5].

Q-5

The study is retrospective or prospective study? Authors said that “we started nutritional intervention for starveling cachectic patients and divided into effective group (n=17) and non-effective group (n=9) using the criteria for cachexia”, a prospective study?

A-5

We apologize for the incomprehensible description.

We performed AHNT for the starveling cachectic patients extracted in the part of 2.1. in the “Materials and methods” section.

Patients who maintained the stage of cachexia for more than three weeks after starting AHNT were considered effective group.

Patients who shifted to the stage of refractory cachexia within three weeks were considered non-effective group.

We retrospectively compared various items between these groups.

Q-6

IRB is lacking.

A-6

This was a retrospective study, and IRB approval was deemed unnecessary.

BY the way, AHNT based on the guideline for parenteral fluid management in terminal cancer patients (edited by JSPM) has become common in Japan, and we performed AHNT for these patients with the approval for performing clinical practice in the hospital.

So we revised the description in the “Ethics“ section as follows:

This study was conducted after obtaining the institutional approval for performing the clinical practice in the hospital and was conducted in accordance with the Declaration of Helsinki.

Round 2

Reviewer 2 Report

Dear Authors,

I appreciate the changes of statistical methods described in the "Statistical analysis" section.

However, it looks like you did not performed the Wilcoxon signed-rank test and the Friedman test because results and p-values are exactly the same. Didn't you find any difference??

In Tables 2 data are still presented as normally distributed. If you perform non-parametric test in the analysis I suppose that variables are not normal. In this case you should present median and interquartile range.  

Did I undestand correctly?

Author Response

Thank you for your reviewing our paper.

Based on the reviewer’s comment, we presented the data using median and interquartile range in the Abstract, the “Results” section of main document, Table 2 and Table 3.

The changes are shown in red and underlined.

Reviewer 4 Report

Thanks for author’s reply.

However, I still have some concerns about the study.

  1. First of all, authors enrolled 223 patients with “refractory cachexia” and identified 26 patients with cachexia with starvation instead of refractory cachexia after ANHT., However, the difference between “cachexia” and “refractory cachexia” is that “low WHO performance status score and a survival period of less than 3 months” (Cancer Management and Research 2020:12 5597–5605) instead of body weight loss or sarcopenia. So, I cannot understand how authors judge these 26 patients to cachexia with starvation from refractory cachexia, live longer than 3 months? It seems not reasonable.
  2. In addition, ANHT were applied in all 233 patients? all received TPN or PPN? TPN and PPN were both not common in terminal patients, As I mentioned before, NG feeding is the most common way for nutrition, the entity of a retrospective study design is not common in hospice care wards. In addition, blood tests on the subsequent 3 weeks nutrition supply, especially CRP and transthyretin, were not routine examined at hospice care wards or in terminal cancer patients? it looks like a prospective study instead of retrospective study. IRB and permits signed by all enrolled patients were still warranted in such a study.

Author Response

Thank you for your reviewing our paper. We revised the manuscript according to the reviewer’s comments. The changes are shown in red and underlined.

Q-1:

First of all, authors enrolled 223 patients with “refractory cachexia” and identified 26 patients with cachexia with starvation instead of refractory cachexia after ANHT., However, the difference between “cachexia” and “refractory cachexia” is that “low WHO performance status score and a survival period of less than 3 months” (Cancer Management and Research 2020:12 5597–5605) instead of body weight loss or sarcopenia. So, I cannot understand how authors judge these 26 patients to cachexia with starvation from refractory cachexia, live longer than 3 months? It seems not reasonable.

A-1:

Sorry for the confusing explanation. The 223 patients selected in 2.1. underwent the necessary interventions, such as symptom management including pain management and peripheral hydration therapy (not AHNT) to improve dehydration, if needed. Then, we again differentiated between patients with refractory cachexia and cachectic patients with starvation. At the time, we used Palliative Performance Scale (PPS) [7] and Palliative Performance Index [8] for the evaluation of performance status and survival time, which were the criteria for diagnosing refractory cachexia.

We revised the relevant part in the “Materials and Methods” section as follows:

The patients selected in 2.1. underwent the necessary interventions, such as symptom management including pain management and peripheral hydration therapy to improve dehydration, if needed. One week after these interventions, we again differentiated between patients with refractory cachexia and cachectic patients with starvation (i.e., patients who were considered to have refractory cachexia at the time of admission, but underwent appropriate symptom relief and hydration, and were subsequently diagnosed as possibly in the cachexia stage) using the criteria for cachexia. At the time, we used Palliative Performance Scale (PPS) [7] and Palliative Performance Index [8] for the evaluation of performance status and survival time, which were the criteria for diagnosing refractory cachexia.

Q-2:

(1) In addition, ANHT were applied in all 233 patients?

(2) all received TPN or PPN? TPN and PPN were both not common in terminal patients,

(3) As I mentioned before, NG feeding is the most common way for nutrition,

(4) In addition, blood tests on the subsequent 3 weeks nutrition supply,

(5) especially CRP and transthyretin, were not routine examined at hospice care wards or in terminal cancer patients? it looks like a prospective study instead of retrospective study. IRB and permits signed by all enrolled patients were still warranted in such a study.

(6) the entity of a retrospective study design is not common in hospice care wards.

A-2:

Sorry for the confusing explanation.              

(1) The intervention on hydration therapy we performed on 223 patients was a peripheral hydration aimed at improving dehydration.

(2) AHNT was performed only for the 26 cachectic patients with starvation.

TPN was mainly performed as AHNT, and when securing a central venous route was difficult, PPN was used.

(3) As the reviewer pointed out, oral intake or enteral nutrition is the most common nutrition supply method for terminal cancer patients, and TPN and PPN were both not common in terminal cancer patients.

As these 26 patients exhibited gastrointestinal symptoms such as nausea, vomiting, appetite loss, and so on (Table 1), AHNT was performed instead of oral nutrition or enteral nutrition via the nasogastric tube.

(4) We do not usually perform regular blood biochemical tests on terminal cancer patients (Blood biochemical tests are done only when physician deems it necessary).

(5) CRP and transthyretin were regularly measured for 26 patients who are taken AHNT to properly evaluate the effectiveness of this intervention (Without this intervention, these items are not measured in terminal cancer patients).

(6) In this study, we analyzed these data retrospectively.

So, we revised relevant part in the “Materials and Methods” section as follows:

As these 26 patients exhibited gastrointestinal symptoms such as nausea, vomiting and appetite loss, AHNT was performed instead of oral nutrition or enteral nutrition via the nasogastric tube. Total parenteral nutrition was mainly performed as AHNT, and when securing a central venous route was difficult, peripheral parenteral nutrition (containing 7.5% glucose and 3% amino acids) was used. Blood biochemical tests including transthyretin were regularly performed to properly evaluate the effectiveness of AHNT.